# The Effect of Subinhibitory Concentration of Metronidazole on the Growth and Biofilm Formation on Toxigenic *Clostridioides difficile* Strains Belonging to Different Ribotypes

**DOI:** 10.3390/pathogens12101244

**Published:** 2023-10-14

**Authors:** Dorota Wultańska, Paweł Karpiński, Michał Piotrowski, Hanna Pituch

**Affiliations:** Department of Medical Microbiology, Medical University of Warsaw, 02-004 Warsaw, Poland; pkarpinski@op.pl (P.K.); piotrowski.michal90@gmail.com (M.P.); hanna.pituch@wum.edu.pl (H.P.)

**Keywords:** *Clostridioides difficile*, PCR-ribotypes, subinhibitory concentration of metronidazole, biofilm

## Abstract

*Clostridioides difficile* is a predominant nosocomial pathogen within the healthcare setting able to produce biofilms. Sub-minimum inhibitory concentrations (sub-MICs) of antibiotics trigger mechanisms affecting bacterial virulence, including increased adhesion and biofilm formation. The aim of this study was to investigate how sub-MICs of metronidazole affect the biofilm formation of *C. difficile* strains. We tested 14 reference and clinical *C. difficile* strains, including hypervirulent strains of RT027. The MICs of metronidazole for the tested strains were determined using the broth microdilution method. Biofilm formation was evaluated using confocal laser scanning microscopy. The *C. difficile* strains belonging to RT027 produced the highest amounts of biofilm. The results of confocal laser scanning microscopy showed that all the tested *C. difficile* strains developed larger biofilms with diversified architectures upon exposure to sub-MICs of metronidazole. In our study, we reveal that sub-MIC concentrations of metronidazole affect the biofilm formation of clinical and reference strains of *C. difficile*. Importantly, metronidazole induces biofilm formation via hypervirulent RT027 strains.

## 1. Introduction

*Clostridioides difficile* is a Gram-positive, obligately anaerobic, spore-forming bacillus that is responsible for a wide spectrum of symptoms, ranging from mild to moderate diarrhea to life-threatening fulminant colitis [1]. Potential sources of *C. difficile* infection (CDI) include asymptomatic carriers, infected patients, hospitals, and animal feces. It is estimated that 5% of adults and 15–75% of newborns may be colonized by *C. difficile*. These proportions significantly increase during hospitalization [1,2]. Hypervirulent strains of *C. difficile* categorized as NAP1/BI/PCR-ribotype 027 (RT027) are a particular threat [3]. Infection with strain RT027 is often characterized by a severe course and increased mortality. It is associated with the overexpression of genes encoding toxins A and B, provoking an increased production of toxins. *C. difficile* strains of the RT027 type are more resistant to antibiotics and produce higher biofilm amounts compared to other RTs [4,5]. In an epidemiological study conducted in Poland in 2011–2013, strains of the RT027 type were isolated from 62% of patients with *C. difficile* infection [6]. The production of toxins is not the sole factor contributing to the pathogenicity and persistence of *C. difficile*; biofilm formation also plays a pivotal role. Biofilms are intricate communities of bacteria residing within an extracellular polymeric substance (EPS) matrix. They are composed of elements such as proteins, polysaccharides, and DNA. These biofilms facilitate *C. difficile* adherence to the gastrointestinal tract lining by providing a stable surface, thereby augmenting their ability to colonize and endure within the host’s gut. This heightened adhesion significantly contributes to *C. difficile*’s overall virulence. Biofilms serve as protective barriers for bacteria, rendering them highly resistant to antibiotic treatments and playing a crucial part in recurrent CDI [7,8].

Metronidazole is a commonly utilized medication in the treatment of various medical conditions. It is recognized for its effectiveness against anaerobic bacterial infections, protozoal infections, and microaerophilic bacterial infections. Moreover, it exhibits cytotoxic properties against facultative anaerobic microorganisms. Metronidazole has gained wide acceptance and official Federal Drug Administration approval for a diverse range of infections, encompassing intestinal amebiases; liver amebiasis; bacterial septicemia; bone and joint infections; central nervous system infections such as meningitis and brain abscesses; endocarditis; gynecologic infections like endometritis, tubo-ovarian abscesses, and bacterial vaginosis; intra-abdominal infections; lower respiratory tract infections; and skin structure infections, and it is used for surgical prophylaxis in colorectal surgeries. For years, metronidazole was used as a first-line therapy for CDI [9,10,11].

When metronidazole enters an organism, it disrupts protein synthesis through its interaction with DNA, ultimately leading to the disruption of helical DNA structure and strand breakage. As a consequence, it triggers cell death in susceptible organisms. The mechanism of action of metronidazole unfolds in a sequential four-step process. The initial step involves its diffusion across the cell membranes of both anaerobic and aerobic pathogens upon entry into an organism, though its antimicrobial effects are specifically targeted toward anaerobes. In the second step, metronidazole undergoes reductive activation facilitated by intracellular transport proteins, modifying the chemical structure of pyruvate-ferredoxin oxidoreductase. This reduction process establishes a concentration gradient within the cell, which facilitates the uptake of more metronidazole and promotes the formation of cytotoxic free radicals. Moving on to the third step, metronidazole’s cytotoxic particles interact with the host cell DNA, resulting in the breakage of DNA strands and the destabilization of the DNA helix, ultimately leading to cell death. Finally, in the fourth step, the cytotoxic products are broken down [8,12,13].

According to the latest recommendations of the European Society of Clinical Microbiology and Infectious Diseases (2021), the recommended drug for the treatment of the first episode of CDI is fidaxomicin, while vancomycin is the second-line drug. If vancomycin and fidaxomicin are not available, the use of metronidazole is acceptable [14].

Sub-minimum inhibitory concentrations (sub-MICs) are concentrations of antibiotics below the MIC value [15]. The results of several studies indicate that sub-MICs of antibiotics induce stress. Also, exposure to sub-MICs of antibiotics triggers different mechanisms affecting physicochemical characteristics and the expression of bacterial virulence factors [16]. One of these mechanisms is increased bacterial adhesion to epithelial cells, which may influence biofilm production via the exposure of bacteria to sub-MICs of an antibiotic [17,18,19]. 

The aim of the present study is to investigate how sub-MICs of metronidazole affect biofilm formation of clinical and reference *C. difficile* strains, including strains belonging to the group of hyper-virulent RTs.

## 2. Materials and Methods

### 2.1. Bacterial Isolates

In the presented study, 14 *C. difficile* strains were analyzed: reference strains ATCC 9689 and *C. difficile* 630 and 12 clinical strains isolated from the stool of patients with diarrhea. Clinical strains belonged to four types of PCR-RTs: 017 (n = 3), 023 (n = 3), 027 (n = 3), and 176 (n = 3). All strains were collected from the Microbial Bank of the Department of Medical Microbiology, Medical University of Warsaw. Tested strains were kept at −70 °C using the Microbank^™^ system (Pro-Lab Diagnostics, Bromborough, Wirral, UK). Before their use, we thawed strains and cultured them on Columbia agar plates with 5% sheep blood (Beckton Dickinson, Franklin Lakes, New Jersey, USA). Strains were incubated in anaerobic conditions using Genbag and Genbox anaer gas generators (bioMérieux, Marcy l’Etoile, France) at 37 °C for 48 h. 

### 2.2. Determination of Minimal Inhibitory Concentration of Metronidazole

We used a method described previously [13] with slight modifications. The MICs of metronidazole for the tested strains were determined using the broth microdilution method in a 96-well plate (Nunc, Roskilde, Denmark). An initial stock solution was prepared by dissolving metronidazole (Sigma-Aldrich, Shanghai, China) in a brain–heart infusion (BHI) medium (BioMaxima S.A., Lublin, Poland). Metronidazole was used at the following concentrations: 0.032, 0.047, 0.064, 0.094, 0.128, 0.256, 0.5, 1, 2, and 4 mg/L. Wells containing 180 μL of dilution were inoculated with 20 μL of *C. difficile* suspension adjusted to 3.0 McFarland and incubated at 37 °C for 48 h under anaerobic conditions. The positive control was BHI medium with 20 μL of bacterial suspension, while the negative control was purely BHI medium. All strains were tested in triplicate to avoid bias. After incubation, optical density at 600 nm was measured using a microplate reader (Bio-Rad, Hercules, CA, USA). Based on the MIC results, we calculated ½ MIC and ¼ MIC values.

### 2.3. Testing the Ability of C. difficile to Produce Biofilm In Vitro

The biofilm assay was performed as described previously [20] with some modifications. The wells of a 96-bottom microplate (Nunc, Roskilde, Denmark) were replenished with 180 μL of BHI medium supplemented with 0.1 M of glucose. Afterward, wells were inoculated with 20 μL of overnight *C. difficile* culture and incubated at 37 °C for 48 h under anaerobic conditions. Wells with BHI broth without inoculum were used as negative controls, while positive controls consisted of inoculated wells. After incubation, the liquid phase was aspired using a sterile pipette, and wells were washed twice with phosphate buffer saline (PBS) (Biomed, Kraków, Poland). Each well was then stained with crystal violet (CV) (Analab, Warsaw, Poland) for 10 min. The CV was removed, and the wells were washed eight times with PBS. After air-drying for 15 min at 37 °C, the CV within the biofilms was dissolved in ethanol, and the absorbance was measured at 620 nm (A620) using a microplate reader. All strains were tested three times. The mean values for each *C. difficile* strain were calculated.

### 2.4. Confocal Laser Microscopy

The samples were visualized using confocal laser scanning microscopy (CLSM) according to the methodology previously described by Waack et al. [21], to which we made certain modifications [22]. We tested reference strains ATCC 9689 and *C. difficile* 630 and one clinical strain per each RT. The biofilm was grown in 2500 µL of BHI supplemented with 0.1 M glucose with 500 µL of overnight *C. difficile* culture on sterile 15 mm diameter glass-bottom dishes (Nunc, Roskilde, Denmark). We used ½ and ¼ concentrations of metronidazole. The positive control was a culture without metronidazole. Dishes were incubated for 48 h at 37 °C under anaerobic conditions. After two days of incubation, the medium was aspired and washed twice with 10 mM of MgSO4. In the next step, the biofilm was stained with acridine orange (10 µg/mL) for 30 min and subsequently washed twice with 10 mM of MgSO4. The stained biofilms were examined using a confocal microscope at the Laboratory of Electron and Confocal Microscopy of the Faculty of Biology, University of Warsaw. The imaging was performed using a Nikon A1R MP microscope based on the Nikon Ti Eclipse biological microscope (Nikon, Tokyo, Japan) under a 60× objective with immersion oil with an image resolution of 2048 × 2048 pixels (0.11 µm/px). An excitation wavelength of 488 nm and an emission wavelength of 500–550 nm were used for reading. The images were processed and analyzed using NIS-Elements AR v. 4.10 software. 

### 2.5. Statistical Analysis

The normal distribution of values was confirmed using the Shapiro–Wilk test. The differences in *C. difficile* biofilm formation between different RTs were calculated using one-way analysis of variance followed by Tukey’s test for multiple comparisons. Analysis was performed using Statistica software (version 13, StatSoft, Warsaw, Poland) and GraphPad Prism (Dotmatics, Boston, MA, USA).

## 3. Results

### 3.1. Minimal Inhibitory Concentration of Metronidazole

The detailed results of the determination of the MICs of the 14 tested *C. difficile* strains are presented in Table 1. The results from each of the three measurements for a single strain were consistent. The lowest MIC values (0.016 mg/L) were observed for two strains belonging to RT023. The highest MIC value (0.5 mg/L) was observed for a strain belonging to RT027. Both reference strains had MICs of 0.094 mg/L. The most prevalent toxigenic profile was TcdA+ TcdB+ CDT+, which was observed in strains belonging to RTs 027, 176, and 023.

### 3.2. The Effect of Sub-Inhibitory Concentrations of Metronidazole on C. difficile Biofilm

The differences in the spontaneous biofilm formation ability of the *C. difficile* strains belonging to the different RTs are presented in Figure 1. The *C. difficile* strains belonging to RT027 produced the highest amounts of biofilm, with an average absorbance of 0.28. This amount was statistically significantly higher than the amounts for the other tested strains (both clinical and reference). Reference strain ATCC produced the lowest amount of biofilm. 

The study of the effect of sub-MICs of metronidazole was repeated for one strain selected from the tested RTs and for both reference strains. We selected the middle strain within the RTs in terms of the amount of biofilm produced in the experiment with CV. Images obtained from confocal laser microscopy revealed the visible effect of sub-MICs of metronidazole on *C. difficile* biofilm (Figure 2).

All the tested *C. difficile* strains developed larger biofilm amounts upon exposure to sub-MICs of metronidazole. However, these biofilms differed depending on the subject strain. Reference strain *C. difficile* 630 (RT012), after exposure to a ½ MIC of metronidazole, produced a thick but sparse biofilm with an irregular biological layer, with a highly distinct 3D architecture containing microaggregates. After exposure to a ¼ MIC, this strain produced a very thick, dense, regular, and highly 3D architecture biofilm containing microaggregates. Reference strain ATCC 9689 (RT001), under the influence of ½ and ¼ MICs of metronidazole, produced a thin, regular biofilm, with no significant differences between the two concentrations of metronidazole, with single, prolonged cells visible in the ½ MIC sample. The control biofilm appeared thin and regular. Clinical strain RT176, after exposure to ½ and ¼ MICs of metronidazole, produced a thick, regular biofilm, but it was denser at a ½ MIC than at a ¼ MIC. The control biofilm was also thick and regular, slightly different from the samples with metronidazole. The clinical RT023 strain produced a notably denser biofilm at a concentration of a ¼ MIC of metronidazole; it was regular, thick, and had a highly 3D architecture and microaggregates, as opposed to the ½ MIC biofilm of metronidazole and the control, which were thinner and has less 3D architectures. The clinical RT017 strain, at a concentration of ¼ MIC of metronidazole, produced a thick, dense biofilm with a highly 3D architecture, but it was irregular compared to the ½ MIC biofilm and the control, which were thinner and irregular. The clinical RT027 strain exposed to a ¼ MIC of metronidazole produced a thick, dense biofilm with a regular 3D architecture. The biofilm of the RT027 strain exposed to a ¼ MIC of metronidazole was thicker than that exposed to a½ MIC of metronidazole and the control. 

## 4. Discussion

Bacterial biofilms represent a critical area of study in the field of medicine, and understanding their significance, especially in the context of pathogens like *C. difficile*, is crucial. Biofilms are complex communities of bacteria encased within a protective matrix, and they have far-reaching implications in healthcare [7]. In the case of *C. difficile*, biofilms hold a key role in the pathogenesis of infections. These biofilms allow *C. difficile* to adhere strongly to the gastrointestinal tract lining, promoting colonization and persistent infection. This heightened adhesion is a major contributor to the severity and recurrence of CDI, placing a significant burden on healthcare systems worldwide. Perhaps even more crucial is the fact that bacterial biofilms, including those formed by *C. difficile*, exhibit heightened resistance to antibiotics [7]. This renders conventional treatment strategies ineffective, leading to prolonged and recurrent infections. In an era where antibiotic resistance is a growing global concern, understanding how biofilms contribute to this issue is vital for developing innovative treatment approaches.

In the study presented herein, our primary objective was to investigate the impact of metronidazole administered at sub-MIC levels on the biofilm formation of *C. difficile*. Remarkably, our observations revealed an increase in the biofilm formation carried out by *C. difficile* strains, including both clinical and reference isolates, when exposed to metronidazole at sub-MIC concentrations. Such an observation shows the harmful effects of antibiotics in low concentrations, and this should certainly be thoroughly investigated. Metronidazole and vancomycin have been the mainstays of CDI treatment for the past few years. The recommendation for metronidazole has been changed due to updated data about drug efficacy [14]. The meta-analysis conducted by Johnson et al. revealed the inferiority of metronidazole compared with vancomycin [23]. Also, the results of a prospective study showed that treatment with metronidazole was a predictor of CDI recurrence [24]. Based on these results, metronidazole is not recommended for the first-line treatment of CDI. 

Strains belonging to the RT027 group are widely recognized as being hypervirulent. In our study, these strains produced larger biofilm amounts compared to strains of other tested RTs. This result confirmed our previous results [5] and those reported by another author [25]. However, sub-MICs of metronidazole increased the biofilm amounts produced by all strains, including the RT027 strain. This fact is of particular concern due to the high rate of strains belonging to the RT027 group in Poland [6]. The significant advantage of this study is our use of clinical strains belonging to different RTs isolated form Polish patients with CDI.

Vuotto et al. [18] demonstrated that sub-MICs of metronidazole increase the production of biofilm by the RT010 strain of *C. difficile* that is susceptible or has reduced susceptibility to metronidazole. The authors suggested that biofilm formation may be one of the components of the multifactorial mechanism of *C. difficile*’s metronidazole resistance. Doan et al. [17] revealed that metronidazole administered in sub-inhibitory concentrations inhibited *C. difficile* motility but also enhanced biofilm production and adherence to Caco-2 cells. In another study performed by the same author [26], the results suggested that the increase in biofilm production could be related to the decrease in the production of the protease Cwp84 and the higher production of a MocR family aminotransferase. The presence of metronidazole may cause a three-fold decrease in the Cwp84 amount [26]. 

Gerber et al. [27] demonstrated that the presence of a sub-inhibitory concentration of metronidazole induces earlier toxin production and influences changes in the gene transcription of the major virulence factors of *C. difficile*.

Other authors have suggested that sub-MICs of antibiotics other than metronidazole can stimulate biofilm formation. Ðapa et al. [28] suggested that sub-MICs of vancomycin also increase biofilm formation by *C. difficile.*

Confocal laser microscopy revealed the visible effect and differences in bacterial morphology. The matrices exposed to sub-MICs of metronidazole were more robust than those of the control. Vuotto et al. [18] suggested that a sub-inhibitory concentration of metronidazole could induce the over-expression of specific genes involved in biofilm production. 

A limitation of our study is the absence of a quantitative evaluation of the impact of metronidazole on *C. difficile* biofilm. We made an attempt to perform this assessment; however, the biofilm cultivated on the microplates exhibited weak adherence to the plastic and was easily washed out. Consequently, we chose to present only the results from CLSM because, unlike CV staining, it enables the observation of a biofilm’s 3D architecture and bacterial cell morphology.

## 5. Conclusions

In our study, we revealed that sub-MIC concentrations of metronidazole affect the biofilm formation of both clinical and reference strains of *C. difficile*. What we observed is interesting: metronidazole, when administered at sub-MIC concentrations, induced biofilm formation across all the tested strains, including the hypervirulent RT027 strain. This allows us to conclude that insufficient antibiotic concentrations can not only fail to eradicate bacteria but may also trigger defense mechanisms of *C. difficile*, such as biofilm formation. However, while our findings shed light on this mechanism, there remains a compelling need for further investigation. More comprehensive studies are required to investigate the underlying mechanisms of this process and thus reveal how sub-MIC concentrations of metronidazole and other antibiotics interact with *C. difficile* to provoke biofilm formation. 

## Figures and Tables

**Figure 1 pathogens-12-01244-f001:**
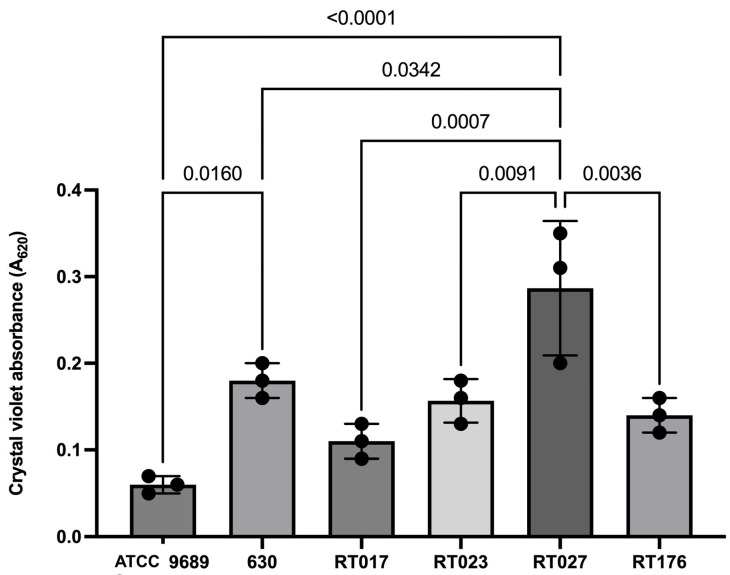
Spontaneous biofilm formation of tested *C. difficile* strains. Data are shown as medians ± standard deviation. Black dots are means from three measurements in ATCC 9689 and 630; in the remaining ribotypes, they are means from three measurements of each strain within this ribotype. In the figure, only statistically significant differences are marked.

**Figure 2 pathogens-12-01244-f002:**
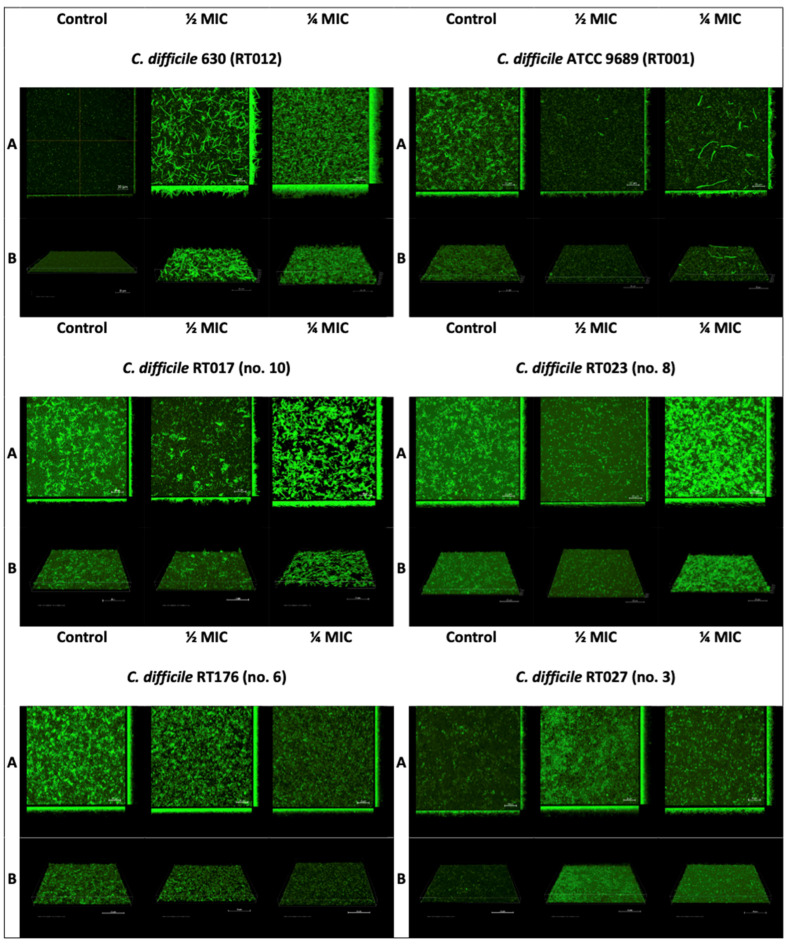
The effect of a sub-inhibitory concentration of metronidazole on *C. difficile* biofilm. (**A**) Projection of the three-dimensional space (x, y, and z) into three two-dimensional planes. (**B**) Diagonal projection of three-dimensional space onto a plane screen.

**Table 1 pathogens-12-01244-t001:** MICs and sub-MICs values of tested *C. difficile* strains.

No. of Strain	Ribotype	Profile of Toxigenicity	MIC (mg/L) Metronidazole
1	027	TcdA+ TcdB+ CDT+	0.094
2	027	TcdA+ TcdB+ CDT+	0.032
3	027	TcdA+ TcdB+ CDT+	0.50
4	176	TcdA+ TcdB+ CDT+	0.032
5	176	TcdA+ TcdB+ CDT+	0.064
6	176	TcdA+ TcdB+ CDT+	0.094
7	023	TcdA+ TcdB+ CDT+	0.032
8	023	TcdA+ TcdB+ CDT+	0.016
9	023	TcdA+ TcdB+ CDT+	0.016
10	017	TcdA− TcdB+ CDT−	0.064
11	017	TcdA− TcdB+ CDT−	0.064
12	017	TcdA− TcdB+ CDT−	0.047
630	012	TcdA+ TcdB+ CDT−	0.094
ATCC 9896	001	TcdA+ TcdB+ CDT−	0.094

CDT—*Clostridioides difficile* binary toxin; MIC—minimal inhibitory concentration; TcdA—*Clostrididioides difficile* toxin A; TcdB—*Clostrididioides difficile* toxin B.

## Data Availability

The data that support the findings of this study are available from the corresponding author upon reasonable request.

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
