# Peer review of "The Effect of Subinhibitory Concentration of Metronidazole on the Growth and Biofilm Formation on Toxigenic Clostridioides difficile Strains Belonging to Different Ribotypes"

_pathogens, 2023, doi:10.3390/pathogens12101244_

Round 1

Reviewer 1 Report (Previous Reviewer 1)

The authors have adequately addressed comments from the previous review.

Author Response

Thank you for this feedback and your time.

Reviewer 2 Report (Previous Reviewer 2)

To inform readers, it would be good to provide toxin information that matches the ribotype of the strain used in table 1.

Sentences that do not conform to grammar, such as lines 93 and 172, require correction.

Author Response

Thank you for this review, and your time. 

To inform readers, it would be good to provide toxin information that matches the ribotype of the strain used in table 1.

Authors: Thank you for this suggestion, we have added profile of toxigenicity.

Sentences that do not conform to grammar, such as lines 93 and 172, require correction.

Authors: Thank you, we have corrected these sentences.

Reviewer 3 Report (Previous Reviewer 3)

Maybe a better choice: replace all “RT” abbreviations, including in the Table 1 by “ribotype” to limit the use of abbreviations. Of course, “RT027” and “RTnnn” remain “RT027” or “RTnnn”

L9 : … settingable … è … setting able …

L20: … metronidatole in induces …è … metronidazole induces…

L21: Clostridoides difficile and L25: Clostridoides difficile: italic or not italic?

L57: In the presented study, we used 14 C. difficile strains: è Fourteen C. difficile strains were analysed:

L134-5: “Black dots are single measurements in ATCC 9689 and 630, in the remaining groups they are means from three measurements.”: there is likely an error: for each triad, there are 3 measurements with the histogram representing the means. Please correct the legend.

L156-8: “Clinical RT017 strain at a concentration of ¼ MIC metronidazole produced thick, dense biofilm with high 3D architecture, but irregular compared to ½  MIC biofilm and control, which were thinner and irregular.”
Where is the description of RT027 (figure n° 3)???

Please, put the order of your figures and the order of their description in the same way!

Author Response

Thank you for this review.

Maybe a better choice: replace all “RT” abbreviations, including in the Table 1 by “ribotype” to limit the use of abbreviations. Of course, “RT027” and “RTnnn” remain “RT027” or “RTnnn”

Authors: Thank you, we have changed it.

L9 : … settingable … è … setting able …

Authors: Thank you, we have corrected it.

L20: … metronidatole in induces …è … metronidazole induces…

Authors: Thank you, we have corrected it.

L21: Clostridoides difficile and L25: Clostridoides difficile: italic or not italic?

Authors: Thank you, we have changed to italic.

L57: In the presented study, we used 14 C. difficile strains: è Fourteen C. difficile strains were analysed:

Authors: Thank you for this suggestion, we have corrected it.

L134-5: “Black dots are single measurements in ATCC 9689 and 630, in the remaining groups they are means from three measurements.”: there is likely an error: for each triad, there are 3 measurements with the histogram representing the means. Please correct the legend.

Authors: We agree, the legend was confusing, and we have improved it.

L156-8: “Clinical RT017 strain at a concentration of ¼ MIC metronidazole produced thick, dense biofilm with high 3D architecture, but irregular compared to ½  MIC biofilm and control, which were thinner and irregular.”
Where is the description of RT027 (figure n° 3)???

Authors: Thank you for this point, we have added description of RT027 strain.

Please, put the order of your figures and the order of their description in the same way!

Authors: Thank you for this suggestion.

Reviewer 4 Report (New Reviewer)

Comments on the documents, including the English language,  can be found in the attached file

Author Response

Thank you for your review and valuable comments. We have addressed all of your comments, and all modifications to the manuscript are marked in track changes mode.

Line 33-34: The correlation between TcdC mutations and overexpression of TcdA and TcdB is not clear. There is a lot of discussion and controversy in the literature about this. The authors should include this, or leave out the emphasis on TcdC (negative regulation gene).

Authors: Thank you, we have leaved this because this is not a subject of this study.

Table 1: Why are the ½ MIC and the ¼ MIC added to this table? This was not measured, but calculated based on the MIC. I think it is impossible to measure ½ MIC or a ¼ MIC.

Authors: Thank you for this point, we have deleted values of ½ and ¼ MIC, and added description to the methods section.

Figure 1. Legend: black dots are single measurements in ATCC 9689 and 630, in the remaining

groups they are means from three measurements.” This is confusing. Does this mean that for the ATCC strain and 630, only one biological replicate was done with three technical replicates measured and for the other strains three biological replicates were done with three technical replicates measured? If so, then the authors should just show that, so that would mean only one black dot for the ATCC strain and 630. And why not test all strains in this assay?

Authors: Authors: We agree, the legend was confusing, and we have improved it.

Figure 2: Very hard to quantify the biofilms in this way. Why not do the crystal violet-based assay for all the strains of the RTs?

Authors: We initially performed a biofilm assay using crystal fillet staining, as in many of our previous publications. Unfortunately, it didn't work out the way we always did, the results were definitely different from those of confocal microscopy. The biofilm as a result of CV staining was not optimal and peeled off easily hence the results were subject to a high risk of bias. Hence, we decided to show the results from confocal microscopy only because they allow us to assess the 3D architecture of the biofilm and the morphology of the bacterial cells.

Why were there three strains per RT tested for MIC, when the other experiments were not carried out with all the strains? I do not understand this. And on what was the selection of each strain based?

Authors: We chosen the middle strain in terms of the amount of biofilm produced in the CV experiment.

Line 170: “In our study, these strains produced larger biofilm amounts compared to strains of other tested RTS._” _

As far as I understood the paper, only one strain per RT was tested for its ability to form biofilms, so why the plural here?

This is repeated in line 201. Please explain or correct me if I am wrong and all RT027 strains were tested.

Authors: Thank you for this point, we have modified this.

Typos/language issues:

Line 9: healthcare settingable to -> healthcare setting, able to

Line 25-27: This sentence does not contain a verb

Line 80: have been -> were

Line 82 culture incubated -> culture and incubated

Line 121: for the strain -> for a strain

Line 129: the highest amounts of biofilm with an average absorbance. This should be followed by a number, like “an average absorbance of 0.28.

Line 131: strain ATCC -> strain ATCC 9689

Line 131: produced the lowest biofilm -> produced the lowest amount of biofilm.

Line 166: the results of the prospective study -> the results of a prospective study

Line 172-173: Although the sub-MICs of metronidazole decreased biofilm amounts produced by all strains including RT027 strains.

This sentence is not correct. Besides, I think it should be increased instead of decreased, since

Lines 141-142 state that All tested C. difficile strains developed larger biofilm amounts upon exposure to sub-MICs of metronidazole” _

Line 177: C. difiicile -> C. difficile

Line 181: in the other study -> in another study

Line 189-191: I think here the same message was written twice. This is a bit awkward. One would expect a better explanation of the findings in the second sentence, but this is just a repeat.

Authors: Thank you for all these points; we have implemented all suggestions into our manuscript.

Round 2

Reviewer 4 Report (New Reviewer)

no comments

no comments

This manuscript is a resubmission of an earlier submission. The following is a list of the peer review reports and author responses from that submission.

Round 1

Reviewer 1 Report

The manuscript is clearly presented and the narrative is easy to follow.  My only significant concern is that it does not really bring anything very novel to the area. A limited number of strains were investigated and most of the data collected was not statistically significant or could not be objectively quantified. These observations could be of interest to those working directly with C. difficile but the data limitations would likely preclude their usefulness in other contexts.

Specific points:

Ln 9- a rather weak opening line for the abstract. A mention of healthcare associated infection would likely give it more impact. Also, the abstract is misleading as it does not mention the lack of statistical significance of the data. This is a significant oversight.

Ln 34 "threat" not "treat"

Ln 47+ An argument around whether sub-MIC levels could be encountered in clinical infection would have been useful as part of a justification for the work. If metronidazole has dropped down the list of therapeutic agents how likely would it be that sub-MIC levels are encountered? 

Ln 74- the number refer to "concentrations" not "dilutions" as the units are mg/L

Ln 155- One strain from each RT was tested for biofilm in the presence of sub-inhibitory MICs.  What criteria were used to select the strain? In some RTs there was not much difference in biofilm formation among the three tested but in others, particularly 027, there were large differences. Picking the highest or lowest biofilm producer might well influence the results.

Some of the data appears contradictory but this is not adequately discussed or explained e.g. Figure 2 shows that sub-MIC levels of metronidazole reduces the amount of biofilm for strain 630. However in Figure 3 the opposite looks to be happening with higher levels of fluorescence in the presence of the drug. Claims are made (Ln 166) that differences in fluorescence for strain RT001 under different conditions are not statistically significant but it is not clear from the methods how this was calculated.

Reviewer 2 Report

This is a biofilm formation study for a total of six ribotypes.

1. MIC study for Metronidazole tested in triplicate for each strain. Please state in Methods how the values presented in Table 1 were calculated.

2. Biofilm assay was performed three times for all strains, and the mean value was calculated and presented. In this case, ATCC9689 and 630 can each have 3 dots, but for the other ribotypes, 9 dots make sure it should be 3 strains * 3 times.

3. In Figures 1 and 2, ATCC 9689 is misspelled as ATTC 9689. Please correct it.

4. Add a description of the abbreviation in Figure 2.

5. Figure 2 and Figure 3 seem to conflict. Except for RT027, it was confirmed that the biofilm analysis value was lower when metronidazole was added than the control group. This appears to conflict with the confocal microscopy presented in Figure 3. It's hard to agree, especially with the RT012 results.

6. Rather than highlighting the results of RT027, it would be more appropriate to conclude that different ribotypes do not have statistical effects on biofilm formation when using metronidazole.

Please check the spelling mistakes.

Reviewer 3 Report

The authors try to compare the biofilm amplitude in C. difficile and its possible dependence on Metronidazole sub-optimal concentration. Biofilm amplitude was measured 1) by crystal violet adsorption  and 2) by confocal 3 D microscopy.

Figure 1: a) it should be clearly indicated that it represents spontaneous biofilms in the absence of antibiotics;

b) comparisons are made between different ribotypes (3 strains for each ribotype), but statistics do not take in account multiple comparisons; 

c) Not a word of explanation for the "small" biofilm of the ATCC strain.

Figure 2: there is no clear evidence of MT on biofilm amplitude in any strain (strain 630 result seems very accidental)

Control in these figures: I guess "Positive controls"? Is should be clearly indicated.

Figure 3: Same questions for "Controls": "Positive controls"?

Some "Controls" in this figure look "negative controls" (strain  012), other "positive controls" (strain 017).

Discussion: while the increased amplitude of biofilm by strain 027 seems visually convincing (despite de statistical inadequate analysis of multiple comparisons), the conclusion of the potential effect of suboptimal doses of MT on increase of biofilm is not at all clearly established. According to your results, suboptimal MT seems to have no effect on amplitude of biofilm.